

# An empirical examination of sample size effects on population demographic estimates in birds using single nucleotide polymorphism (SNP) data

Jessica F. McLaughlin[1,2] and Kevin Winker[1]

[1] University of Alaska Museum & Department of Biology and Wildlife, University of Alaska Fairbanks, Fairbanks, AK, USA
[2] Sam Noble Oklahoma Museum of Natural History and Department of Biology, University of Oklahoma, Norman, OK, USA

Corresponding author
Kevin Winker,
kevin.winker@alaska.edu

## ABSTRACT

Sample size is a critical aspect of study design in population genomics research, yet few empirical studies have examined the impacts of small sample sizes. We used datasets from eight diverging bird lineages to make pairwise comparisons at different levels of taxonomic divergence (populations, subspecies, and species). Our data are from loci linked to ultraconserved elements and our analyses used one single nucleotide polymorphism per locus. All individuals were genotyped at all loci, effectively doubling sample size for coalescent analyses. We estimated population demographic parameters (effective population size, migration rate, and time since divergence) in a coalescent framework using Diffusion Approximation for Demographic Inference, an allele frequency spectrum method. Using divergence-with-gene-flow models optimized with full datasets, we subsampled at sequentially smaller sample sizes from full datasets of 6–8 diploid individuals per population (with both alleles called) down to 1:1, and then we compared estimates and their changes in accuracy. Accuracy was strongly affected by sample size, with considerable differences among estimated parameters and among lineages. Effective population size parameters ($v$) tended to be underestimated at low sample sizes (fewer than three diploid individuals per population, or 6:6 haplotypes in coalescent terms). Migration ($m$) was fairly consistently estimated until <2 individuals per population, and no consistent trend of over-or underestimation was found in either time since divergence ($T$) or theta ($\Theta = 4N_{ref}\mu$). Lineages that were taxonomically recognized above the population level (subspecies and species pairs; that is, deeper divergences) tended to have lower variation in scaled root mean square error of parameter estimation at smaller sample sizes than population-level divergences, and many parameters were estimated accurately down to three diploid individuals per population. Shallower divergence levels (i.e., populations) often required at least five individuals per population for reliable demographic inferences using this approach. Although divergence levels might be unknown at the outset of study design, our results provide a framework for planning appropriate sampling and for interpreting results if smaller sample sizes must be used.

# INTRODUCTION

Genomic-scale data for studying population histories have increased the resolution of demographic estimates, including effective population sizes, migration rates, and times since divergence, even when the number of sampled individuals is relatively low (*Willing, Dreyer & Van Oosterhout, 2012*; *Jeffries et al., 2016*; *Nazareno et al., 2017*). However, it is not well understood how the precision and accuracy of these estimates are impacted by lower population sample sizes. The number of individuals that can be included in a study might be limited by practical considerations such as availability of samples for difficult-to-access or endangered populations, tradeoffs between including more individuals per population or more populations, or decisions about whether to include more loci or more individuals (*Felsenstein, 2005*; *Pruett & Winker, 2008*; *Jeffries et al., 2016*). Because these issues affect study design, it is important to understand the impacts of relatively low within-population sample sizes on population demographic parameters that are now commonly estimated in a coalescent framework.

The impacts of population sample size, and particularly the tradeoff between increased numbers of individuals versus increased number of loci, has been studied primarily with microsatellite datasets. In general, increasing the number of loci decreases the number of individuals needed for accurate parameter estimations in population genetic studies (*Morin, Martien & Taylor, 2009*; *Willing, Dreyer & Van Oosterhout, 2012*), but different parameter estimates are not impacted uniformly by low sample sizes. A sample size of eight alleles per population (4:4 diploid individuals) has been suggested as an optimum sample size for obtaining coalescent-based likelihood estimates of theta ($\Theta = 4N_e\mu$; *Felsenstein, 2005*). This sample size has also been sufficient for non-coalescent-based estimates of unbiased heterozygosity (*Pruett & Winker, 2008*), which have been effectively estimated with 5–10 individuals. However, other estimators, such as genetic diversity (e.g., $A_E$, $H_O$ and unbiased $H_E$) and differentiation ($F_{ST}$), require larger sample sizes for accurate estimation, and often the number of individuals required increases as divergence decreases (*Kalinowski, 2005*; *Morin, Martien & Taylor, 2009*).

Modern genomic datasets, with their large numbers of sampled loci, are predicted to decrease the number of individuals required for obtaining accurate estimates of demographic history (*Jeffries et al., 2016*). However, impacts of sample size on such estimates have undergone only limited investigation thus far, and previous empirical work has focused on estimates of diversity ($A_E$, $H_O$ and unbiased $H_E$) and differentiation ($F_{ST}$; *Nazareno et al., 2017*). Other demographic estimates made using allele frequency spectrum methods have only been evaluated so far with simulated data (*Robinson et al., 2014*), using the program Diffusion Approximation for Demographic Inference (δaδi; *Gutenkunst et al., 2009*). *Robinson et al. (2014)* showed that median estimated parameter values in two-population δaδi models of divergence in isolation remained close to true values down to three diploid individuals per population. However, this did not hold true

across all three model types they examined, and their optimal sampling recommendations depended on the timescale of the demographic events experienced by the populations, with very recent and very ancient events both requiring greater sample sizes (*Robinson et al., 2014*). In empirical systems, such information on the timescale of demographic events or divergence might be unknown at the outset of a study, particularly in taxa that have not been previously studied, and care must be taken to avoid sampling too few individuals to accurately estimate parameters of interest.

Here we use empirical datasets to conduct pairwise examinations of how inferences of population parameters are impacted by sample size, scaling symmetrically downwards from full datasets that meet or exceed sample sizes widely considered optimal for coalescent-based analyses. We expected that as sample sizes decreased, errors in estimates would increase and accuracy would decrease, but to varying degrees among parameters, and that systematic biases of mean estimates of parameters might emerge at lower sample sizes. We used empirical datasets from diverging avian lineages with different demographic and evolutionary histories to enhance our understanding of how lower sample sizes affect estimates of effective population size ($v$), migration ($m$), time since divergence ($T$), and $\Theta$ ($4N_{ref}\mu$).

# MATERIALS AND METHODS

## Study system

We used eight datasets of ultraconserved elements (UCEs) from Beringian birds from *McLaughlin et al. (2020*; Table 1). Genomic data were generated using the methods outlined in *Winker et al. (2019)* using protocols from Dr. Travis Glenn's lab at the University of Georgia (http://baddna.uga.edu/protocols.html). From these data, we generated repeatedly subsampled datasets at smaller sample sizes for analysis under a coalescent framework using δαδi (*Gutenkunst et al., 2009*). While we tried other programs, we were unable to get them to consistently run on our UCE datasets despite months of effort and over a hundred thousand hours of high-performance computing resources (e.g., jaatha, *Mathew et al., 2013*; IMa2p, *Sethuraman & Hey, 2015*). Thus, although we lack independent corroboration, we consider δαδi to be sufficient to answer the questions posed. Our empirical datasets represent taxonomically designated levels of population, subspecies and species pairs in three avian orders, contrasting pairs of Asian and North American populations of: *Clangula hyemalis* (long-tailed duck), *Anas crecca crecca/A. c. carolinensis* (green-winged teal), and *Mareca penelope/M. americana* (Eurasian and American wigeons) in Anseriformes; *Numenius phaeopus variegatus/N. p. hudsonicus* (whimbrel) and *Tringa brevipes/T. incana* (gray-tailed and wandering tattlers) in Charadriiformes; and *Luscinia svecica* (bluethroat), *Pinicola enucleator kamschatkensis/ P. e. flammula* (pine grosbeak) and *Pica pica/P. hudsonia* (Eurasian and black-billed magpies) in Passeriformes. These datasets, which span divergence levels from populations with substantial levels of gene flow to effectively reproductively isolated species (albeit with low gene flow), enable us to explore how the effects of low sample sizes on demographic inference play out across these levels of divergence. Insofar as taxonomy is not a reliable indicator of genomic divergence levels (*Humphries & Winker, 2011*), we also

**Table 1 Datasets and divergence.**

|  | Variable loci | Full dataset size | $F_{ST}$ |
|---|---|---|---|
| Anseriformes |  |  |  |
| *Clangula hyemalis* | 2,442 | 7:7 | 0.004 |
| *Anas crecca* | 2,481 | 6:6 | 0.02 |
| *Mareca penelope /M. americana* | 2,315 | 8:8 | 0.044 |
| Charadriiformes |  |  |  |
| *Numenius phaeopus* | 2,388 | 7:7 | 0.269 |
| *Tringa brevipes /T. incana* | 1,636 | 8:8 | 0.585 |
| Passeriformes |  |  |  |
| *Luscinia svecica* | 2,516 | 7:7 | 0.014 |
| *Pinicola enucleator* | 2,656 | 7:7 | 0.442 |
| *Pica pica/Pica hudsonia* | 2,199 | 7:7 | 0.328 |

Note:
Number of variable loci in each lineage, the full dataset size (number of diploid individuals in each population), and $F_{ST}$ values (from *McLaughlin et al. (2020)*).

include in our evaluations estimates of $F_{ST}$ made from the full datasets (Table 1). Among the lineages in this study, pairwise comparisons fell out into two general groups, one with relatively low divergence and one with relatively high divergence (*McLaughlin et al., 2020*; Table 1).

These datasets consist of one single nucleotide polymorphism (SNP) per locus from over 1,500 UCE loci per lineage (each lineage is a pairwise, two-population sample of diverging populations, subspecies, or species). For bioinformatics methods, see *Winker et al. (2019)* and *McLaughlin et al. (2020)*; a summary of our pipeline is given here: https://github.com/jfmclaughlin92/beringia_scripts. Each dataset consists of 100% coverage for all individuals (all individuals have phased, high-quality SNPs called at both alleles for all loci). Z-linked loci were removed because they have a different inheritance scalar from autosomal loci (*Winker et al., 2019*; *McLaughlin et al., 2020*). Original sequence data are deposited in the NCBI Sequence Read Archive (SRA; PRJNA393740). Complete data files analyzed for this study are available at https://doi.org/10.6084/m9.figshare. 12622658.v1.

## Subsampling datasets and analyses

To produce datasets of varying sample sizes, stepping down from the maximum number of individuals available for each population (6–8) to 1 individual per population, a custom Python script (https://github.com/jfmclaughlin92/beringia_scripts) was used. This script (ngapi_dadi.py) iteratively sampled individuals without replacement from the thinned .vcf files, created new .vcf files containing these individuals, converted these files to the proper δaδi input format (using a Perl script by Kun Wang, https://groups.google. com/forum/#!msg/dadi-user/p1WvTKRI9_0/1yQtcKqamPcJ), and ran δaδi models with predetermined, lineage-specific best-fit parameters for the split-migration (divergence-with-gene-flow) model that comes with the δaδi Demographics2D.py file (split-mig). For six of our eight lineages, split-migration models produced a best-fit

model among multiple options, while for two of them a secondary contact model was a demonstrably better fit (*Clangula hyemalis* and *Mareca penelope/americana*; *McLaughlin et al., 2020*). Here we chose to include all eight datasets under a single model framework (split-migration, an isolation-with-migration model in δaδi, termed "split-mig" therein). We wished to focus here on changes due to sample size variation with multiple empirical datasets and not on more subtle variation due to differences among divergence-with-gene-flow models.

For each sample size, 25 subsampled datasets were created, which were each run five times. The best-fit run by highest maximum log composite likelihood value among those five runs was then selected for each dataset and used for subsequent analyses. Parameter estimates for effective population size ($v_1$ and $v_2$), migration ($m$), divergence time ($T$), and Θ (defined as $4N_{ref}\mu$, with $N_{ref}$ defined as ancestral population size and μ as mutation rate per generation), were then compared across different sample sizes. Raw parameter estimates are used throughout; we did not convert these values to individuals or years (except for three illustrative examples for individuals; see below) because that would introduce lineage-specific idiosyncrasies (e.g., through application of different mutation rate estimates) that would diminish the power of our focal among-lineage comparisons here. For the three exemplar cases in which we translated raw values into numbers of individuals, we used estimates of mutation rate and generation time given in Table S2.

The scaled root mean square error (SRMSE) was calculated, defined as

$$SRMSE_\theta = \frac{\sqrt{\frac{\Sigma\left(\hat{\theta} - \theta\right)^2}{n}}}{\bar{\theta}}$$

with θ in this context representing the estimate from the full dataset, $\hat{\theta}$ as the parameter estimate from the subsampled dataset, and $n$ the number of datasets (25) considered, following *Robinson et al. (2014)*. This was scaled by the mean of the parameter estimate at each sample size ($\bar{\theta}$) to enable inter-lineage comparisons of the changes in accuracy at lower sample sizes (SRMSE). This allowed us to quantify the changes in accuracy of estimates at different sample sizes relative to each species' parameter estimates' means.

## RESULTS

Each lineage had a dataset of between 1,636 and 2,656 variable loci (Table 1). Across the eight lineages, 25 datasets were constructed at each sample size from 1:1 individual up to the full sample size minus one for a total of 1,250 subsampled datasets.

Overall, as expected, variability in parameter estimates increased and accuracy decreased with smaller sample sizes (Table 2; Fig. 1; Figs. S1–S5). Performance of mean parameter estimates varied both with lineage and with sample size. The effective population size parameters ($v_1$ and $v_2$) tended to be underestimated at the lowest sample sizes, whereas there was a trend towards overestimation of migration at the lowest sample sizes ($m$; Table 2; Fig. 1; Figs. S1–S5). Divergence time ($T$) and Θ were more ambiguous, with both

**Table 2 Demographic parameter estimates.**

| | Parameter | 8:8 | 7:7 | 6:6 | 5:5 | 4:4 | 3:3 | 2:2 | 1:1 |
|---|---|---|---|---|---|---|---|---|---|
| **Anseriformes** | | | | | | | | | |
| *Clangula hyemalis* | $v_1$ | – | 8.937 (±1.068) | 10.706 (±0.449) | 11.039 (±0.327) | 10.662 (±0.319) | 10.977 (±0.234) | 10.688 (±0.275) | 8.864 (±0.532) |
| | $v_2$ | – | 6.410 (±1.012) | 10.704 (±0.255) | 10.657 (±0.318) | 10.634 (±0.388) | 11.546 (±0.130) | 9.915 (±0.344) | 9.851 (±0.525) |
| | $T$ | – | 1.487 (±0.213) | 1.542 (±0.065) | 1.460 (±0.067) | 1.497 (±0.083) | 1.472 (±0.053) | 1.639 (±0.105) | 2.155 (±0.187) |
| | $m$ | – | 1.217 (±0.229) | 1.524 (±0.121) | 1.554 (±0.137) | 1.704 (±0.148) | 1.847 (±0.143) | 2.093 (±0.190) | 2.324 (±0.157) |
| | $\Theta$ | – | 204.806 (±33.285) | 136.062 (±4.133) | 140.407 (±6.721) | 139.646 (±6.591) | 133.497 (±2.999) | 129.653 (±4.928) | 116.837 (±6.160) |
| *Anas crecca* | $v_1$ | – | – | 13.529 (±0.268) | 13.515 (±0.229) | 13.801 (±0.380) | 12.598 (±0.516) | 13.261 (±0.526) | 11.129 (±0.722) |
| | $v_2$ | – | – | 16.737 (±0.450) | 16.689 (±0.471) | 16.523 (±0.492) | 16.939 (±0.516) | 15.270 (±1.061) | 11.631 (±1.090) |
| | $T$ | – | – | 1.154 (±0.039) | 1.226 (±0.019) | 1.265 (±0.024) | 1.333 (±0.045) | 1.298 (±0.046) | 1.500 (±0.088) |
| | $m$ | – | – | 0.736 (±0.063) | 0.83 (±0.040) | 0.661 (±0.073) | 0.699 (±0.114) | 0.472 (±0.094) | 0.765 (±0.147) |
| | $\Theta$ | – | – | 143.00 (±4.157) | 135.50 (±1.492) | 133.17 (±1.581) | 130.67 (±3.204) | 133.51 (±2.831) | 127.13 (±5.231) |
| *Mareca penelope/ M. americana* | $v_1$ | 10.116 (±0.002) | 10.518 (±0.132) | 9.847 (±0.318) | 10.063 (±0.217) | 9.904 (±0.260) | 9.398 (±0.320) | 10.193 (±0.466) | 9.438 (±0.618) |
| | $v_2$ | 15.608 (±0.004) | 15.147 (±0.192) | 14.895 (±0.237) | 14.531 (±0.334) | 14.082 (±0.302) | 14.015 (±0.535) | 12.644 (±0.562) | 6.276 (±0.713) |
| | $T$ | 1.139 (±0.000) | 1.135 (±0.023) | 1.209 (±0.035) | 1.190 (±0.021) | 1.214 (±0.023) | 1.235 (±0.036) | 1.268 (±0.043) | 1.267 (±0.077) |
| | $m$ | 0.704 (±0.000) | 0.750 (±0.095) | 0.644 (±0.021) | 0.654 (±0.028) | 0.716 (±0.049) | 0.529 (±0.062) | 0.568 (±0.093) | 1.761 (±0.247) |
| | $\Theta$ | 128.06 (±0.012) | 128.83 (±1.794) | 125.16 (±1.024) | 125.08 (±1.178) | 123.56 (±1.177) | 123.76 (±1.912) | 121.81 (±1.946) | 128.64 (±4.026) |
| **Charadriiformes** | | | | | | | | | |
| *Numenius phaeopus* | $v_1$ | – | 2.982 (±0.003) | 2.887 (±0.021) | 2.845 (±0.029) | 2.722 (±0.030) | 2.614 (±0.043) | 2.332 (±0.051) | 2.542 (±0.138) |
| | $v_2$ | – | 6.245 (±0.004) | 6.086 (±0.066) | 6.047 (±0.064) | 5.691 (±0.085) | 5.308 (±0.097) | 4.735 (±0.127) | 4.176 (±0.211) |
| | $T$ | – | 1.968 (±0.002) | 1.931 (±0.019) | 1.981 (±0.040) | 1.894 (±0.040) | 1.796 (±0.063) | 1.501 (±0.052) | 2.386 (±0.132) |
| | $m$ | – | 0.056 (±0.000) | 0.055 (±0.001) | 0.056 (±0.001) | 0.052 (±0.003) | 0.042 (±0.004) | 0.023 (±0.007) | 0.133 (±0.013) |
| | $\Theta$ | – | 147.88 (±0.104) | 149.67 (±1.009) | 147.32 (±1.271) | 150.84 (±1.960) | 157.11 (±3.098) | 173.13 (±3.298) | 141.10 (±6.502) |
| *Tringa brevipes/ T. incana* | $v_1$ | 7.894 (±0.093) | 8.487 (±0.093) | 7.516 (±0.166) | 7.014 (±0.166) | 6.382 (±0.267) | 5.258 (±0.625) | 2.806 (±0.292) | 1.016 (±0.086) |
| | $v_2$ | 2.559 (±0.045) | 2.835 (±0.036) | 2.663 (±0.055) | 2.537 (±0.085) | 2.613 (±0.103) | 2.395 (±0.111) | 1.416 (±0.150) | 0.578 (±0.050) |
| | $T$ | 6.575 (±0.134) | 7.624 (±0.107) | 7.284 (±0.189) | 7.153 (±0.291) | 7.542 (±0.364) | 7.033 (±0.389) | 3.856 (±0.536) | 1.942 (±0.203) |
| | $m$ | 0.0091 (±0.000) | 0.0081 (±0.000) | 0.0084 (±0.000) | 0.0085 (±0.000) | 0.0090 (±0.000) | 0.0098 (±0.000) | 0.008 (±0.002) | 0.165 (±0.015) |
| | $\Theta$ | 56.345 (±0.986) | 49.828 (±0.628) | 52.707 (±1.250) | 54.627 (±2.022) | 53.799 (±4.510) | 58.978 (±4.510) | 113.161 (±10.291) | 117.030 (±8.657) |
| **Passeriformes** | | | | | | | | | |
| *Luscinia svecica* | $v_1$ | – | 3.877 (±0.005) | 3.934 (±0.089) | 4.618 (±0.344) | 5.056 (±0.408) | 5.827 (±0.435) | 6.322 (±0.488) | 5.452 (±0.403) |
| | $v_2$ | – | 21.452 (±0.092) | 20.980 (±0.442) | 18.847 (±0.961) | 15.954 (±1.072) | 15.795 (±1.156) | 14.675 (±1.307) | 15.969 (±1.432) |
| | $T$ | – | 1.290 (±0.003) | 1.285 (±0.015) | 1.276 (±0.031) | 1.243 (±0.063) | 1.226 (±0.043) | 1.203 (±0.067) | 1.256 (±0.104) |
| | $m$ | – | 1.956 (±0.058) | 2.122 (±0.108) | 2.127 (±0.245) | 2.330 (±0.347) | 3.357 (±0.334) | 2.416 (±0.332) | 2.940 (±0.312) |
| | $\Theta$ | – | 166.94 (±0.176) | 167.608 (±1.033) | 167.935 (±2.405) | 176.558 (±5.360) | 172.299 (±3.636) | 175.525 (±4.166) | 180.675 (±8.488) |

| | Parameter | 8:8 | 7:7 | 6:6 | 5:5 | 4:4 | 3:3 | 2:2 | 1:1 |
|---|---|---|---|---|---|---|---|---|---|
| *Pinicola enucleator* | $v_1$ | – | 2.519 (±0.016) | 2.846 (±0.057) | 2.843 (±0.076) | 2.658 (±0.113) | 2.597 (±0.120) | 2.197 (±0.121) | 2.325 (±0.117) |
| | $v_2$ | – | 1.786 (±0.011) | 2.355 (±0.013) | 2.112 (±0.046) | 1.898 (±0.063) | 1.656 (±0.050) | 1.412 (±0.037) | 1.465 (±0.073) |
| | $T$ | – | 1.979 (±0.021) | 2.449 (±0.028) | 2.317 (±0.076) | 2.098 (±0.099) | 1.866 (±0.077) | 1.568 (±0.048) | 2.480 (±0.184) |
| | $m$ | – | 0.0073 (±0.001) | 0.0105 (±0.000) | 0.0107 (±0.001) | 0.00677 (±0.001) | 0.0033 (±0.001) | 0.0010 (±0.001) | 0.0596 (±0.004) |
| | $\Theta$ | – | 223.76 (±1.51) | 197.10 (±1.41) | 205.45 (±3.30) | 219.25 (±5.23) | 233.07 (±5.22) | 256.52 (±4.80) | 212.34 (±11.87) |
| *Pica pica/Pica hudsonia* | $v_1$ | – | 2.699 (±0.042) | 2.485 (±0.046) | 2.406 (±0.057) | 2.298 (±0.075) | 2.300 (±0.094) | 2.117 (±0.142) | 1.567 (±0.144) |
| | $v_2$ | – | 7.107 (±0.126) | 6.759 (±0.225) | 6.470 (±0.330) | 6.604 (±0.390) | 6.565 (±0.501) | 5.537 (±0.528) | 3.029 (±0.587) |
| | $T$ | – | 3.334 (±0.069) | 3.017 (±0.046) | 2.868 (±0.067) | 2.710 (±0.089) | 2.561 (±0.114) | 2.325 (±0.143) | 2.309 (±0.190) |
| | $m$ | – | 0.0141 (±0.000) | 0.0121 (±0.000) | 0.0119 (±0.010) | 0.0086 (±0.001) | 0.0066 (±0.001) | 0.0033 (±0.001) | 0.0808 (±0.012) |
| | $\Theta$ | – | 108.09 (±1.602) | 116.50 (±1.48) | 121.01 (±2.00) | 126.62 (±3.26) | 132.95 (±4.74) | 146.85 (±7.58) | 162.11 (±9.71) |

**Note:**

Mean estimates (± SEM) of effective population size parameters ($v_1$ and $v_2$), migration ($m$), time since divergence ($T$), and $\Theta$ (defined as $4N_{ref}\mu$, where $N_{ref}$ is ancestral population size and $\mu$ is mutation rate per generation), in eight lineages of trans-Beringian birds calculated from 25 resampled datasets at each sample size.

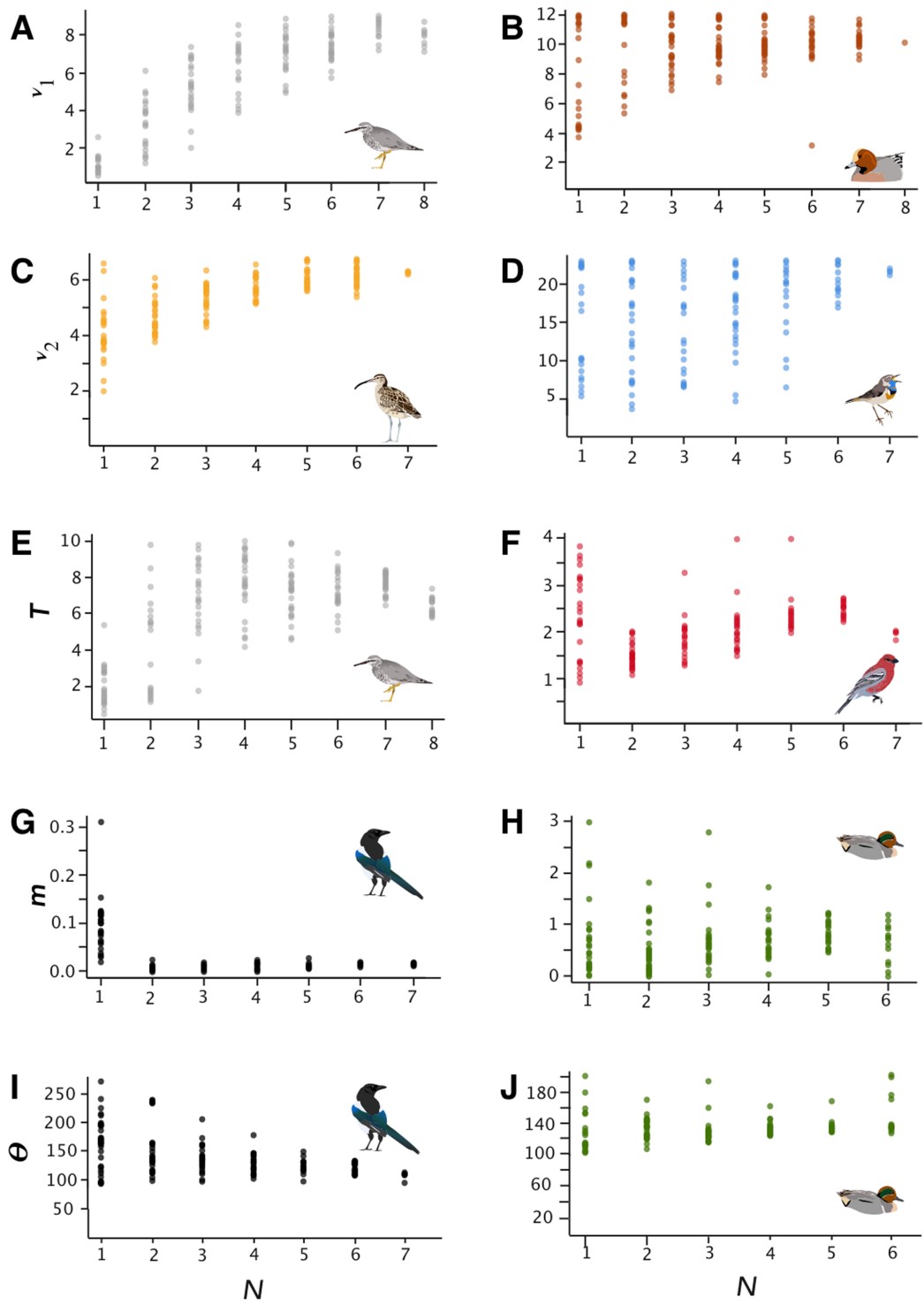

**Figure 1 Parameter estimates.** Parameter estimates of effective population size ($\nu_1$ and $\nu_2$), time since divergence ($T$), migration ($m$), and $\Theta$ for selected lineages (parameters are raw, unconverted values directly from δαδι analyses). Taxa are *Tringa brevipes/incana* (A, E), *Mareca penelope/M. americana* (B), *Numenius phaeopus* (C), *Luscinia svecica* (D), *Pinicola enucleator* (F), *Pica pica/P. hudsonia* (G, I), and *Anas crecca* (H, J).

**Table 3 Scaled root mean square errors.**

| | Parameter | 7:7 | 6:6 | 5:5 | 4:4 | 3:3 | 2:2 | 1:1 |
|---|---|---|---|---|---|---|---|---|
| | | 7 | 6 | 5 | 4 | 3 | 2 | 1 |
| *Clangula hyemalis* | $m$ | | 0.391 | 0.438 | 0.451 | 0.436 | 0.540 | 0.500 |
| *Anas crecca* | $m$ | | | 0.262 | 0.556 | 0.803 | 1.123 | 0.942 |
| *Mareca penelope/M. americana* | $m$ | 0.624 | 0.187 | 0.221 | 0.339 | 0.661 | 0.838 | 0.913 |
| *Numenius phaeopus* | $m$ | | 0.082 | 0.128 | 0.330 | 0.627 | 2.026 | 0.753 |
| *Tringa brevipes/T. incana* | $m$ | 1.293 | 1.213 | 1.214 | 1.081 | 0.928 | 1.659 | 0.992 |
| *Luscinia svecica* | $m$ | | 0.262 | 0.570 | 0.747 | 0.642 | 0.699 | 0.619 |
| *Pinicola enucleator* | $m$ | | 2.809 | 2.779 | 5.003 | 11.108 | 38.261 | 0.486 |
| *Pica pica/Pica hudsonia* | $m$ | | 0.650 | 0.771 | 1.459 | 2.182 | 5.303 | 1.053 |
| | | 7 | 6 | 5 | 4 | 3 | 2 | 1 |
| *Clangula hyemalis* | $\nu_1$ | | 0.225 | 0.156 | 0.174 | 0.122 | 0.156 | 0.432 |
| *Anas crecca* | $\nu_1$ | | | 0.081 | 0.136 | 0.214 | 0.195 | 0.384 |
| *Mareca penelope/M. americana* | $\nu_1$ | 0.073 | 0.161 | 0.106 | 0.130 | 0.183 | 0.224 | 0.329 |
| *Numenius phaeopus* | $\nu_1$ | | 0.049 | 0.069 | 0.109 | 0.163 | 0.299 | 0.318 |
| *Tringa brevipes/T. incana* | $\nu_1$ | 0.116 | 0.109 | 0.178 | 0.282 | 0.512 | 1.788 | 6.511 |
| *Luscinia svecica* | $\nu_1$ | | 0.112 | 0.398 | 0.459 | 0.496 | 0.541 | 0.464 |
| *Pinicola enucleator* | $\nu_1$ | | 0.141 | 0.167 | 0.275 | 0.307 | 0.505 | 0.427 |
| *Pica pica/Pica hudsonia* | $\nu_1$ | | 0.157 | 0.202 | 0.271 | 0.296 | 0.462 | 0.907 |
| | | 7 | 6 | 5 | 4 | 3 | 2 | 1 |
| *Clangula hyemalis* | $\nu_2$ | | 0.141 | 0.164 | 0.193 | 0.156 | 0.170 | 0.261 |
| *Anas crecca* | $\nu_2$ | | | 0.138 | 0.146 | 0.150 | 0.354 | 0.635 |
| *Mareca penelope/M. americana* | $\nu_2$ | 0.069 | 0.091 | 0.135 | 0.151 | 0.217 | 0.320 | 1.588 |
| *Numenius phaeopus* | $\nu_2$ | | 0.059 | 0.061 | 0.122 | 0.198 | 0.345 | 0.554 |
| *Tringa brevipes/T. incana* | $\nu_2$ | 0.154 | 0.133 | 0.168 | 0.204 | 0.228 | 0.887 | 3.241 |
| *Luscinia svecica* | $\nu_2$ | | 0.106 | 0.285 | 0.477 | 0.507 | 0.635 | 0.557 |
| *Pinicola enucleator* | $\nu_2$ | | 0.094 | 0.109 | 0.210 | 0.332 | 0.537 | 0.526 |
| *Pica pica/Pica hudsonia* | $\nu_2$ | | 0.182 | 0.282 | 0.308 | 0.391 | 0.566 | 1.702 |
| | | 7 | 6 | 5 | 4 | 3 | 2 | 1 |
| *Clangula hyemalis* | $T$ | | 0.329 | 0.310 | 0.358 | 0.282 | 0.434 | 0.632 |
| *Anas crecca* | $T$ | | | 0.095 | 0.128 | 0.213 | 0.207 | 0.369 |
| *Mareca penelope/M. americana* | $T$ | 0.100 | 0.153 | 0.097 | 0.111 | 0.164 | 0.194 | 0.315 |
| *Numenius phaeopus* | $T$ | | 0.052 | 0.066 | 0.110 | 0.196 | 0.350 | 0.322 |
| *Tringa brevipes/T. incana* | $T$ | 0.138 | 0.149 | 0.208 | 0.261 | 0.275 | 1.006 | 2.508 |
| *Luscinia svecica* | $T$ | | 0.056 | 0.118 | 0.250 | 0.178 | 0.284 | 0.408 |
| *Pinicola enucleator* | $T$ | | 0.183 | 0.291 | 0.437 | 0.578 | 0.848 | 0.396 |
| *Pica pica/Pica hudsonia* | $T$ | | 0.193 | 0.265 | 0.350 | 0.444 | 0.608 | 0.673 |
| | | 7 | 6 | 5 | 4 | 3 | 2 | 1 |
| *Clangula hyemalis* | $\Theta$ | | 0.262 | 0.295 | 0.296 | 0.264 | 0.333 | 0.490 |
| *Anas crecca* | $\Theta$ | | | 0.077 | 0.094 | 0.153 | 0.126 | 0.237 |
| *Mareca penelope/M. americana* | $\Theta$ | 0.068 | 0.046 | 0.052 | 0.059 | 0.083 | 0.094 | 0.153 |
| *Numenius phaeopus* | $\Theta$ | | 0.035 | 0.042 | 0.067 | 0.113 | 0.173 | 0.231 |

(Continued)

| Table 3 (continued) | | | | | | | | |
| --- | --- | --- | --- | --- | --- | --- | --- | --- |
| | Parameter | 7:7 | 6:6 | 5:5 | 4:4 | 3:3 | 2:2 | 1:1 |
| *Tringa brevipes/T. incana* | $\Theta$ | 0.337 | 0.283 | 0.280 | 0.338 | 0.395 | 0.608 | 0.670 |
| *Luscinia svecica* | $\Theta$ | | 0.030 | 0.070 | 0.158 | 0.108 | 0.126 | 0.242 |
| *Pinicola enucleator* | $\Theta$ | | 0.060 | 0.118 | 0.186 | 0.224 | 0.284 | 0.298 |
| *Pica pica/Pica hudsonia* | $\Theta$ | | 0.123 | 0.161 | 0.218 | 0.278 | 0.386 | 0.463 |

**Note:**
Scaled root mean square error (SRMSE) for each parameter at each diminished sample size. Parameters are effective population size ($v_1$ and $v_2$), migration ($m$), time since divergence ($T$), and $\Theta$ (defined as $4N_{ref}\mu$, where $N_{ref}$ is ancestral population size and $\mu$ is mutation rate per generation).

over-and under-estimation occurring in different lineages (Table 2; Fig. 1; Figs. S1–S5). These corresponded in many cases to large changes in the biologically meaningful estimates derived from these parameters. For example, this can be seen in the effective population size parameter of *Tringa brevipes* ($v_1$), which varied from 1.02 to 8.49 across the full sample size spectrum (Table 2). This represents effective population size estimates of 4,478 to 37,410 individuals. In other cases, however, seemingly large changes translated into minor biological differences (e.g., changes in $m$ among pairwise comparisons with very low levels of gene flow, considered in more detail below).

In general, SRMSE increased as sample sizes decreased (Table 3; Fig. 2), reflecting the loss of accuracy at lower sample sizes. Lineages with lower levels of divergence (Table 1; Fig. S6) tended to exhibit more variability among model runs at higher sample sizes than lineages with higher levels of divergence (e.g., *Numenius* vs. *Luscinia* in Fig. 1 for $v_2$). This was most notable in the two population-level splits (*L. svecica* and *C. hyemalis*; Fig. 1; Figs. S1–S5). At higher levels of divergence (Table 1)—particularly among *T. brevipes/T. incana*, *N. phaeopus*, and *Pica pica/Pica hudsonia*—most parameter estimates reached a consistent level at approximately 4 or 5 diploid individuals, after which adding more individuals did not considerably improve estimates (Table 2), whereas SRMSE generally only began to increase markedly below 3:3 comparisons for population size and split-time estimates (Table 3; Fig. 2). In some lower-divergence lineages, such as *A. crecca* and *L. svecica*, SRMSE began increasing substantially in most parameters below a sample size of 5 (Table 3; Fig. 2). However, this was not universally the case, with SRMSE values in *C. hyemalis* remaining similar at most sample sizes for multiple parameter estimates (Table 3; Fig. 2).

Variation among lineages was noteworthy, as was variation among demographic variables as sample sizes changed. Considering aggregate performance, using SRMSE as the basis for among-lineage contrasts, all lineages showed a significant decrease in performance (increased SRMSE) with smaller sample sizes (Table 4). These relationships were all significant using a linear regression except for the SRMSE of $m$, which showed aberrancies at $N = 2$ among some high-divergence lineages (Tables 2–4; Fig. 2; Fig. S6). In many cases the linear regression models were substantially improved by breaking the lineages into low-divergence and high-divergence groups (groups from Table 1, split by $F_{ST}$ values < 0.05 and >0.25; Table S1).

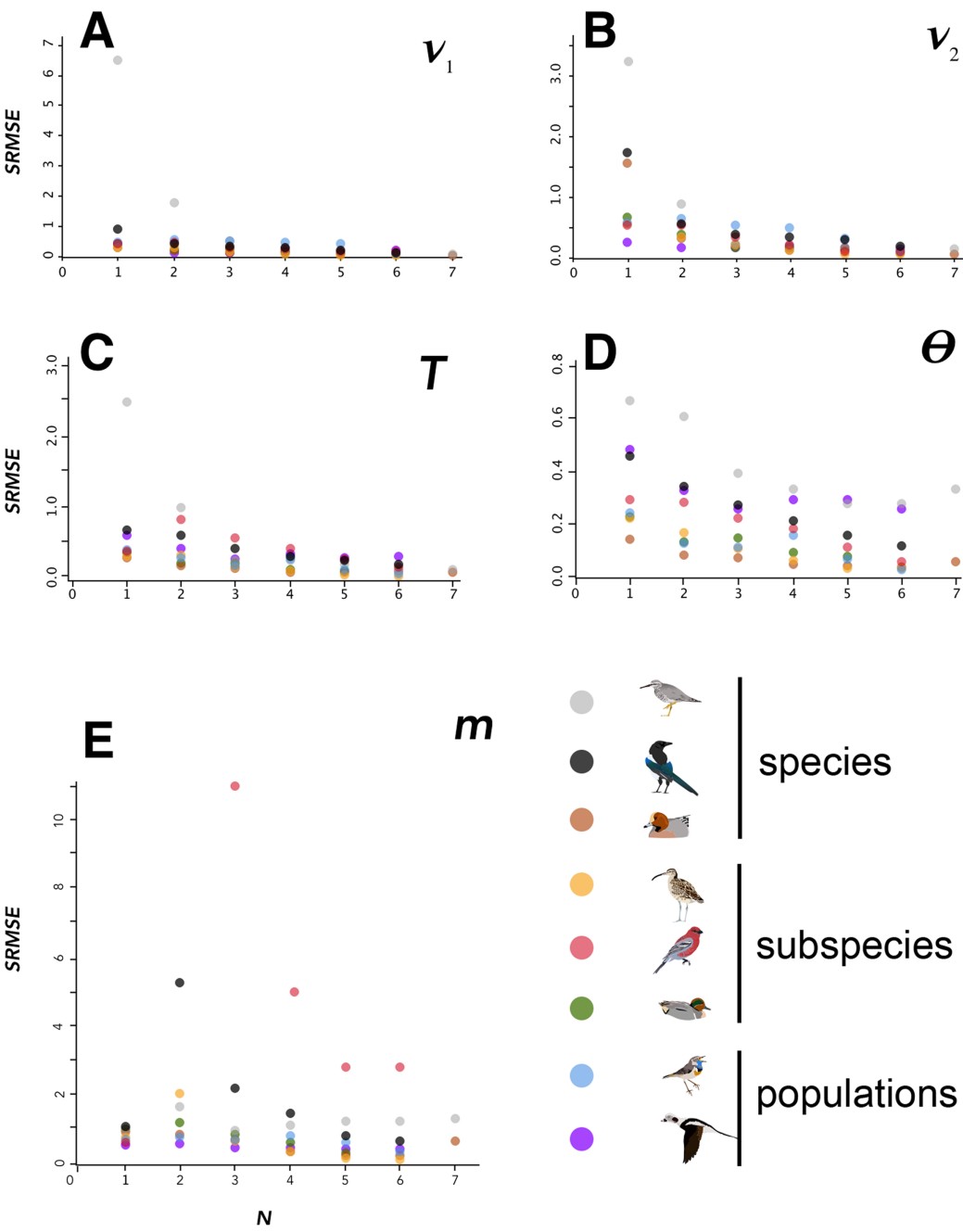

**Figure 2 SRMSE values.** SRMSE values for demographic parameters estimated at various sample sizes in this study, indicating how estimates decrease in accuracy with smaller sample sizes. Pairwise comparisons within each lineage are coded at lower right. Note that vertical scales are different in each panel. A–E indicate each of the demographic parameters estimated in this study, given at the upper right in each panel.

## DISCUSSION

Sample size is an important consideration in study design, but it remains understudied in large-scale genomic datasets (*Nazareno et al., 2017*). Our results suggest that the minimum reliable sample size will vary considerably from taxon to taxon, depending on factors

**Table 4 Regressions.** Linear regression equations for scaled root mean square error (SRMSE) for each parameter (from Table 3), summarizing how accuracy declines with diminished sample sizes. Parameters are migration ($m$), effective population size ($v_1$ and $v_2$), time since divergence ($T$), and $\Theta$ (defined as $4N_{ref}\mu$, where $N_{ref}$ is ancestral population size and $\mu$ is mutation rate per generation). Note that these are based on SMRSE values (to enable among-lineage comparisons). Thus, $y$ in the regression equation $y = mx + b$ is SMRSE for that particular demographic variable ($m$ is slope, $x$ is $N$ and $b$ is the $y$ intercept).

| SMRSE for variable | Regression equation ($y = mx + b$) | $r^2$ | $P$ |
|---|---|---|---|
| $m$[a] | $y = -0.00858 \times N + 0.87714$ | 0.413 | 0.0007 |
| $v_1$ | $y = -0.17563 \times N + 1.05063$ | 0.117 | 0.0156 |
| $v_2$ | $y = -0.16171 \times N + 0.97351$ | 0.305 | 0.00004 |
| $T$ | $y = -0.10045 \times N + 0.70147$ | 0.237 | 0.0004 |
| $\Theta$ | $y = -0.03825 \times N + 0.34738$ | 0.220 | 0.0007 |

Note:
[a] Note that this is for the low-divergence group only.

such as parameters of interest and the depth of the lineage's divergence. Although analyses using coalescent theory have suggested that sample sizes of 8–10 individuals per population are optimal (*Felsenstein, 2005*), by genotyping both alleles of diploid animals our sample sizes were doubled (i.e., $1N = 2$ haplotypes), and we were able to estimate population parameters at considerably lower sample sizes in terms of individuals. Certain parameters, such as migration rate ($m$) and effective population sizes ($v_1$ and $v_2$), showed fairly consistent patterns of bias in over-or under-estimation across all lineages (Fig. 1; Figs. S1–S5). In particular, gene flow ($m$) was fairly consistently estimated with relatively small departures from accuracy down to two individuals per population, after which it was overestimated in all lineages (Table 2; Fig. S3).

## Estimates of migration

We found the most variation in estimates of $m$ occuring when samples were at 2:2 (e.g., *Pinicola enucleator* and *Pica pica/hudsonia*; Fig. 2). In most of the cases in which extreme estimates occurred at 2:2, pairings of individuals that caused geographic clustering of within-continent population samples were involved together with numerically very small estimates of $m$. The values of $m$ were consistently small, but variation around the mean estimate was apparently magnified by more subtle within-continent variation than our study was designed to detect. Biologically, we reason that small values of $m$ are the more informative takeaway, and that increased variation around those very small numbers at $N$ of 2:2 is an artifact arising from a combination of relatively deep divergence and very low gene flow, probably coupled with some more subtle population structure within continental populations. In biological terms, although these variations can appear graphically substantial (Fig. 2, $m$), in *Pinicola enucleator* they represented estimates ranging (max–min) from 0.01 to $6.13 \times 10^{-9}$ individuals per generation. In *Pica*, these max–min values were 0.03–$2.29 \times 10^{-9}$ individuals per generation.

## Estimates of population size

The effective population sizes ($v$ parameters) were not as robust, with variation tending to begin to increase markedly below four diploid individuals per population and accuracy

decreasing in all lineages (Tables 2 and 3; Fig. 1; Figs. S1 and S2). They were, however, still reasonably accurate in many lineages at relatively small samples sizes (Tables 2 and 3; Fig. 1; Figs. S1 and S2). The negative relationships between SRMSE values for each demographic parameter and sample size ($N$) should help users interpret how lineages and individual parameters are affected by smaller sample sizes (Table 4; Table S1).

## The impact of divergence

Our results reinforce previous findings (*Kalinowski, 2005*; *Morin, Martien & Taylor, 2009*) that an important factor in determining the minimum sample size for a study is the level of divergence in the lineages under examination. Although this might be known at the start of a study, that might not always be true, potentially complicating sampling design. However, some general recommendations are possible, at least within a broader framework of higher-and lower-divergence groups. Lineages with considerable divergence (e.g., species-level, such as in *Tringa*) had accurate demographic parameters estimated at lower sample sizes (Fig. 2; Figs. S1–S5). Thus, it seems possible in such systems to reliably use fewer individuals. In shallowly diverged populations that might experience substantial gene flow, however, higher sample sizes may be required to overcome the impact of individuals with varying amounts of admixture, which appears to increase the variation in model performance at lower sample sizes among low-divergence lineages (Fig. 1; Figs. S1–S6; Table S1).

Our findings of the effects of divergence levels on the minimum sample sizes needed to accurately estimate population demographic parameters broadly agreed with previous findings in other genetic markers, with some exceptions. In lineages that are more shallowly split and have experienced more gene flow, greater sample sizes are required to reliably estimate multiple parameters, including not just the demographic parameters examined here, but also genetic distance (*Kalinowski, 2005*), $F_{ST}$ (*Morin, Martien & Taylor, 2009*; *Humphries & Winker, 2011*), and recent demographic events (e.g., <100 generations; *Beichman, Huerta-Sanchez & Lohmueller, 2018*). The two population-level splits in our study, *L. svecica* and *C. hyemalis*, did not perform as well for most parameter estimates at sample sizes below 6 individuals per population, with accuracy (as measured by SRMSE; Table 3) decreasing rapidly; this fits our understanding that accurately estimating more recent demographic events requires the improved draw on more recent coalescent events that increased sample sizes bring (*Beichman, Huerta-Sanchez & Lohmueller, 2018*). The presence of a substantial amount of gene flow appears to increase variation in parameter estimates and decrease accuracy, as seen in *L. svecica* (Tables 2 and 3), and in practical terms would require increased sample sizes for accurate parameter estimation.

## Model fit

Due to computational restrictions, we analyzed all subsampled datasets under the split-mig δaδi (*Gutenkunst et al., 2009*) model determined and optimized for the full dataset in each lineage, and we did not investigate the impact of sample size on model fit. Several subsample datasets (notably *Clangula hyemalis*, *Mareca penelope/M. americana*, and *Luscinia svecica*) showed signs in some parameters of beginning to consistently

push the upper bounds of some model parameters. This means that both variation and over-estimation of the parameters were likely underestimated in these groups at smaller sample sizes. This situation has also been noted with simulated data, which have been found in some situations to have a better fit with a model type different than the one under which they were simulated (*Robinson et al., 2014*).

**Implications for study design**

Research efficiency requires attention not only to the minimum sample size required to meet an objective, but also to the point after which adding more samples begins to produce diminishing returns. In this context, this means the point above which the SRMSE becomes similar between sample sizes, but before the means of estimates start to change due to decreased sample size. This inflection point might represent the minimum reliable sample size, but not necessarily. In some lineages, SRMSE was very similar at larger sample sizes, began to slowly increase at intermediate sizes, and then at low sample sizes increased quickly (Table 3; Fig. 2). This again varied among lineages (Table 3; Figs. S1–S5). In some, such as the *Pica* and *Tringa* species lineages, this inflection point was reached at higher sample sizes than the minimum reliable sample sizes in some parameters (Table 3), whereas in others, such as in most estimates of *m*, these points were the same (e.g., Fig. S3). However, in some groups, particularly estimates of effective population size ($v_1$) and migration (*m*) in *L. svecica*, this optimal point was not reached until the full dataset was analyzed, and might not have been reached at all in *C. hyemalis* in any of the parameter estimates (Figs. S1–S5). This is consistent with the findings of *Robinson et al. (2014)*, in that although in some cases a small sample size could be used, larger sample sizes still led to more accurate parameter estimates. This was especially the case in our data for divergence times (*T*), Θ, and some effective population size (*v*) estimates (Table 2; Fig. 1; Figs. S1–S5). Our linear regression models help generalize these relationships (Table 4; Table S1). In sum, two key sources of variation preclude our providing detailed suggestions for threshold sample sizes in future studies: the levels of divergence in a study's focal lineage, and the demographic parameter of most interest for that study. We urge those designing their own studies to consider our results (Tables 2–4, Supplemental Information) at different divergence depths and for different demographic parameters, depending on objectives.

## CONCLUSIONS

Sample size is a critical aspect of study design and interpretation, and balancing the need for reliable estimates with cost effectiveness is a key tradeoff. Inadequate sampling can lead to ambiguous or biased results (*Nazareno & Jump, 2012*; *Nazareno et al., 2017*), whereas many parameter estimates are not improved above a certain sample size (*Felsenstein, 2005*; *Nazareno et al., 2017*). As other researchers, we found that inference of demographic parameters can be strongly influenced by sample size, with estimates becoming less accurate at lower sample sizes and being over- and underestimated, with considerable variation both among parameters and among lineages. In general, for

pairwise comparisons at shallow levels of divergence (population), care should be taken to include adequate samples, with the best performance in these data generally occurring at 6 or more diploid individuals per population. Parameter estimates in lineages with deeper divergence (subspecies and species) were generally more resilient to lower sample sizes.

## ACKNOWLEDGEMENTS

The University of Washington Burke Museum provided some specimens for use in this study. Python scripts by Kevin Hawkins were a vital resource in constructing the custom scripts used in this study. Thanks also to Kathryn Everson, Alexandra Lewis, Naoki Takebayashi, Kris Hundertmark, and two anonymous reviewers for comments on earlier drafts.

### Funding

The authors received no funding for this work.

### Competing Interests

The authors declare that they have no competing interests.

### Author Contributions

- Jessica F. McLaughlin conceived and designed the experiments, performed the experiments, analyzed the data, prepared figures and/or tables, authored or reviewed drafts of the paper, and approved the final draft.
- Kevin Winker conceived and designed the experiments, prepared figures and/or tables, authored or reviewed drafts of the paper, and approved the final draft.

### DNA Deposition

The following information was supplied regarding the deposition of DNA sequences:
Data are available at NCBI SRA: PRJNA393740.

### Data Availability

The scripts are available at GitHub:
https://github.com/jfmclaughlin92/beringia_scripts
The data are available at figshare: Winker, Kevin; McLaughlin, Jessica (2020):
Sample size effects on population demographic estimates in birds using single nucleotide polymorphism (SNP) data. figshare. Dataset. DOI 10.6084/m9.figshare.12622658.v1.

### Supplemental Information

Supplemental information for this article can be found online at http://dx.doi.org/10.7717/peerj.9939#supplemental-information.

Peer

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
