# Peer review of "An empirical examination of sample size effects on population demographic estimates in birds using single nucleotide polymorphism (SNP) data"

_PeerJ, doi:10.7717/peerj.9939_

## Round 0.1 · original submission · Major Revisions

Both reviewers saw merit in your paper. After a careful read, I also think it should be eventually published after major revisions, because it approaches an issue little explored in the literature, despite its great importance. I pointed out my main concerns below:

1. avoid including citations and too many acronyms in the abstract
2. spell out that is theta the first time you use the symbol in the introduction
3. Reviewer 1 pointed out about the transferability of the study setting and I'd like to elaborate on that by asking you to improve the reproducibility of the analysis. Because this is a methodological paper, readers would benefit from having detailed instructions. Since you partially used Python scripts, I highly recommend you to set up a Jupyter online notebook describing your analytical workflow. Also, improve your reasons why you focused only on one software. Is not there any other that could provide the same analysis?
4. Both Reviewers point out the overall emphasis on an unpublished paper and the lack of details about data collection and generation. I second them and I'd recommend you to use protocols.io to describe all your lab procedures in more detail.
5. in the topic Implications for Study Design in the discussion, try to improve the interpretation of the regression models and give more specific guidance about thresholds in numbers of individuals sampled.

Reviewer 1 ·

Basic reporting

NA

Experimental design

NA

Validity of the findings

NA

Additional comments

This study addresses an issue of quite wide interest: what sample size is needed to obtain reliable estimates of population demographic estimates? This is studied by subsampling from empirical data, rather than pure simulation. The study is based on SNP recovered from ultraconserved elements in birds. They used resampling to construct simulated populations with a random subset of individuals to access accuracy in population demographic parameters. The system is valuable for addressing the questions posed and the topic will be of interest to many readers of PeerJ.

Although the study has its merits, it is very much focused on a specific setup that might not be transferable to other species and studies. The authors need to investigate the effect of the number of SNPs on the population demographic parameters (i.e., what the minimum number of SNPs to obtain accurate estimates?). In addition, to properly assess whether the data set used is appropriate for the analyses presented, the authors need to provide more information about the SNP data, since the article cited (McLaughlin et al. 2020) has not yet been published. For instance, what does the minor allele frequency distribution look like in the populations? If this is skewed, it can also affect the estimate of population demographic parameters. Another feature of the data, that I would find interesting to check is whether it would be possible to get phased data per sequenced UCE locus and therefore more than two alleles. Would the estimates of the statistics benefit from it?

Authors also need to justify why they, specifically, used the program δaδi (Diffusion Approximation for Demographic Inference; Gutenkunst et al. 2009). It may be very interesting to see if different approaches can affect the accuracy of population demographic parameters when small sample sizes are used.

Reviewer 2 ·

Basic reporting

McLaughlin and Winker test the effects of varying sample size in estimating demographic parameters using dadi. The study is very well-written, interesting, and certain to be of broad interest. I have very few problems overall with this study and am recommending only minor revisions.

My main issues is that the methods are not adequately described in my opinion. For example, there is not any information about sequencing or library prep protocol or extraction. Similarly, there is no information about bioinformatics. How were SNPs called? How were UCEs assembled? and so on?

I understand that these data are primarily from another study, but that study appears not to be published yet. Not only that, but Peer J’s policy is for published papers to be self-contained. While this lack of self-containment has a slight stench of "salami science" (i.e. chopping up research to produced more papers), I do think it is self-contained enough to warrant publication. Therefore, to fit the journal's standard on self-containment, the authors should at a minimum provide basic information about labwork, sequencing, and bioinformatics, with a citation to the other paper. However, I would rather the authors keep it completely self-contained, and describe these protocols in detail. Much of this can be presented as supplementary data if there are major space limitations.

One additional point is that the units are not stated for any parameter anywhere as far as I could tell. Not in figures, not in methods, not in results. What are the units of effective population size (v1 and v2) and divergence time (T)? Is divergence time in years? Thousands? Millions of years? Even though this study does not focus on these values per se, they are necessary to avoid any confusion and are of interest to the reader.

The assumed mutation rate is also not provided.

Overall, though, I think the paper is a great addition to literature and will be cited by many.

Experimental design

This study meets all the journal's standards on experimental design.

Validity of the findings

The findings are quite interesting and will be useful for other researchers desiging studies with current generation approaches such as UCEs.

Additional comments

Some minor points:

Line 153: the number of loci is stated once in the Materials and Methods. Best to not be redundant. Pick one to use. I personally prefer results.

Line 161: “Time since split” is a bit awkward, how about simply using “divergence time” throughout.

Figure 1: Ideally, I would really like to see these plots for each species and each parameter. A table is really not the ideal way to display these results. I see they are in supplemental, but if you can put this in the main text, it would improve the paper.

Also, throughout, you call the models "split-migration" models. In my experience many/most researchers refer to these as isolation-with-migration models. It is more precise/informative to use the this term in my opinion. "split-migration" is a bit ambiguous without more detail.

---

## Round 0.2 · Minor Revisions

I believe authors have indeed improved their manuscript by making it a "stand-alone text", which could be read without direct reference to the other paper (which is now cited as being in press in Mol Ecol). They also responded to each critique in a fair manner and did the best as they could. I think Figs S1-S5 are ok as given in the suppl mat and Fig. 1 in the main text, so no need to change that. The Methods are much more clearer now, with added links to protocols and data processing pipeline.

I just would like to kindly ask you to deal with the issue on alternative software use both reviewers touched upon and resubmit a revised version.

Reviewer 1 ·

Basic reporting

NA

Experimental design

NA

Validity of the findings

NA

Additional comments

I thank the author for addressing many of my concerns, and I find the manuscript improved. However,
authors need to give a more clear justification of why they only used the Diffusion Approximation for Demographic Inference. What are you mean with 'we were unable to get them to consistently run on our UCE datasets'?

Reviewer 2 ·

Basic reporting

The paper I think is much improved from its earlier version.

Specifically, the methods are now clear and explicitly stated.

Experimental design

This study meets all the journal's standards on experimental design.

Validity of the findings

Excellent

Additional comments

Some minor comments:

Line 93-95: "While we tried other programs, we were unable to get them to consistently run on our UCE datasets (e.g., jaatha, Mathew et 95 al. 2013; IMa2p, Sethuraman & Hey 2015)." I know you added this sentence for Referee #1, but it just seems like a lame excuse for not using additional software. I personally feel that dadi is sufficient to answer the question you've asked, and so I would simply drop this line. This is ultimately up to you, but I feel that readers will not sympathize with "this is possibly important but we couldn't get it to work."

---

## Round 0.3 · accepted · Accept

Thank you for addressing the critiques on alternative software. I would gladly recommend acceptance of this revised version.